# Factors affecting yawning frequencies in preterm neonates

**Damiano Menin**[1], **Elisa Ballardini**[2], **Roberta Panebianco**[1], **Giampaolo Garani**[3], **Caterina Borgna-Pignatti**[2], **Harriet Oster**[4], **Marco Dondi** [1]*

**1** Dipartimento di Studi Umanistici, Università degli Studi di Ferrara, Ferrara, Italy, **2** Department of Medical Sciences, University of Ferrara, Ferrara, Italy, **3** Azienda Ospedaliera-Universitaria, Ferrara, Italy, **4** School of Professional Studies, New York University, New York City, New York, United States of America

* marco.dondi@unife.it

## Abstract

Yawning is a long neglected behavioral pattern, but it has recently gained an increasing interdisciplinary attention for its theoretical implications as well as for its potential use as a clinical marker, with particular regard to perinatal neurobehavioral assessment. The present study investigated the factors affecting yawning frequencies in hospitalized preterm neonates (N = 58), in order to distinguish the effects of hunger and sleep-related modulations and to examine the possible impact of demographic and clinical variables on yawning frequencies. Results showed that preterm neonates yawned more often before than after feeding, and this modulation was not explained by the amount of time spent in quiet sleep in the two conditions. Moreover, second born twins, known to be more prone to neonatal mortality and morbidity, showed increased yawning rates compared to first born twins. Overall, our results are consistent with the hypothesis that yawning frequencies in preterm neonates are modulated by separate mechanisms, related e.g. to hunger, vigilance and stress. These findings, although preliminary and based only on behavioral data, might indicate that several distinct neuropharmacological pathways that have been found to be involved in yawn modulation in adults are already observable in preterm neonates.

## Introduction

Yawning is a phylogenetically conserved behavior, virtually ubiquitous in vertebrates [1] that can be observed in isolation or in bursts, often accompanied by stretching of the upper limbs [2]. This behavioral pattern has been long neglected by scholars, but recently it has been gaining increasing interdisciplinary attention [3]. This interest is partly due to its peculiar position at the crossroads of very different phenomena. In fact, yawning frequencies in humans have been found to be modulated by different conditions, including hunger [4, 5], arousal and circadian rhythms [6, 7], thermoregulation [8], pain [9] and stress [10]. Moreover some neurological pathologies [11] as well as the intake of specific drugs [12, 13] can result in variations in yawning rates. This behavior is also contagious in humans as well as in other strongly social species, e.g. apes [14, 15] and dogs [16, 17], and this fact has led some scholars to consider

**Competing interests:** The authors have declared that no competing interests exist.

yawning as a potential window into the origins of motor contagion and social interaction [3]. During the last decades, several theories have been formulated to explain the evolutionary origins of yawning, each identifying different modulating factors as the core function of this behavioral pattern, including arousal/vigilance [1, 18], brain thermoregulation [19], cortisol levels regulation [20], empathy and social interaction [3, 21]. Despite this tendency to trace the origins of yawning back to a single function, recent studies have highlighted in some primates a surprising morphological and temporal variability in yawns, associated with different conditions and social contexts [22–24], suggesting that yawning might serve different functions in different circumstances [24].On the other hand, although these theories have been often presented as competing explanatory alternatives, several scholars have highlighted the need of distinguishing proximate and ultimate explanations when investigating mechanisms underlying the manifestation of yawning behavior [25]. In particular, advocates of the brain cooling hypothesis have argued that, while, e.g., empathy, familiarity or communication might serve as proximal mechanisms explaining yawning modulation, the ultimate function(s) of yawning is likely physiological and should be able to explain proximate mechanisms as well [26].

Yawning is also an ontogenetically primitive behavior, observed in human fetuses from the eleventh gestational week [18, 27] as well as in preterm and full-term neonates [28, 29]. However, very few studies have investigated the dynamics of yawning modulation in early development [28, 30, 31]. The study of yawning in fetuses and preterm neonates is particularly relevant to the investigation into the ultimate function of yawning, not only because it makes it possible to investigate the ontogenetically primary functions of yawning, but also because the very existence of fetal yawning has been proposed as evidence against the brain cooling hypothesis, based on the fact that the mother controls thermoregulation of the fetus [32]. Gallup and Eldakar [33], however, have argued that yawning, similarly to other behavioral patterns, might serve different functions (if any) during prenatal life than after birth. With regard to this hypothesis, studying preterm neonates (i.e. born before having completed the 37th week of gestation) could allow us to distinguish maturational and environmental aspects of the early functional development of yawning.

Moreover, as yawning is related to different homeostatic processes [7], detailed analysis of yawning may also help to identify potential applications to neurobehavioral assessment [34, 35].

The present study aims to investigate the factors affecting yawning frequencies in hospitalized preterm neonates, in order to distinguish the effects of hunger and sleep-related modulations and to explore the possible impact of demographic or clinical variables on yawning frequencies.

In particular, since hunger has proven to modulate spontaneous behavior (including hand-mouth coordination) in full-term neonates and infants [36–38], we asked whether a similar trend can be established for yawns, by observing the spontaneous behavior of preterm neonates before and after feeding. If confirmed, the increase in yawning rates prior to feeding would be consistent with a role in brain thermoregulation, as there is evidence, although limited, that feeding in preterm neonates is integrated into a heat production episode [39–40]. Prior animal research, in fact, has shown that yawns are triggered during rises in brain/body temperature [41] and result in a reduction of facial temperatures [42]. Moreover, the frequency of yawning increases during rising ambient temperature and diminishes at low ambient temperatures [43]. Similarly, warming of the carotid arteries increases yawning while cooling of this blood flow to the brain decreases yawning [44].

A secondary question concerned the degree to which any modulation observed before vs. after feeding could be ascribed to differences in behavioral states. In fact, preliminary observations indicate that yawning is generally absent during quiet sleep (QS), both in neonates [28,

30] and fetuses [35]. Evidence of the effects of other behavioral states is limited. Moreover, quiet sleep (QS) has been found to be the only behavioral state that shows stability in the perinatal period, and can be identified more reliably than active or REM sleep, especially when scored based on behavioral analysis alone [45, 46].

By scoring QS periods, easily identifiable even without relying on physiological data, we were therefore able to examine the potential effect of the total time spent in QS on yawning frequencies in preterm neonates, and to investigate whether this sleep-related modulation accounts for the difference between yawning before and after feeding. Moreover, in order to investigate whether the hypothesized difference in yawning rates before and after being fed might be partly or completely due to a shift in behavioral state distributions, further analyses were conducted after excluding QS periods from the considered observation time.

Additionally, the effects of demographic and clinical variables were assessed in order to identify possible confounders and pinpoint developmental trends or morbidity-dependent yawn modulation. In particular, the inclusion of a group of twins allowed us to test the difference between the first and the second born. The second twin has in fact been found to be more prone to neonatal mortality and morbidity, probably because of the increased risk of hypoxia during delivery [47, 48]. Second born twins may therefore be expected to display higher frequencies of stress-related yawning [9, 10]. Moreover, developmental trajectories of preterm neonates (i.e., those born before 37 completed weeks of gestation) are to be understood based on the interaction of three separate age variables, i.e., gestational age (GA, the duration of the gestation), chronological age (CH, the time from birth to observation) and postmenstrual age (PMA, the sum GA and CH) [49]. As a consequence, we tested the potential associations between each of these three variables, as well as gender, and yawning frequencies. Finally, because promising evidence points at yawn duration as a potential marker of different types of yawns in some non-human primates [22–24], we investigated potential differences in yawn durations before and after feeding.

## Methods

### Participants

The study sample consisted of 58 appropriate for gestational age (AGA, i.e. whose birth weight was between the 10th and 90th percentiles for the infant's gestational age and sex), preterm neonates (26 males and 32 females) born between 24 and 36 weeks of gestational age (GA, M = 32.53; SD = 3.01) and observed between 32 and 42 weeks postmenstrual age (PMA, M = 35.60; SD = 1.51), including 30 singletons and 14 twin pairs (see Table 1 for detailed demographics). Exclusion criteria were: congenital anomalies, heart or metabolic disorders, fetal infections, clear teratogenic factors, Apgar at five minutes < 6 and grade III or IV hemorrhages. This study was carried out in strict accordance with the recommendations outlined by the American Psychological Association and the Italian Association of Academic Psychologists

**Table 1. Sample demographics.**

| Group | n | GA (weeks) | CH (days) | PMA (weeks |
|---|---|---|---|---|
| Twins | 28 | 33.17 (2.96) | 17.43 (19.15) | 35.67 (1.25) |
| Singletons | 30 | 31.94 (2.98) | 25.20 (20.60) | 35.54 (1.73) |
| Total | 58 | 32.53 (3.01) | 21.45 (20.16) | 35.60 (1.51) |

Values are expressed as Mean (SD); GA, Gestational Age; CH, Chronological Age

PMA, Postmenstrual Age.

and the study was approved by the Ethics Committee of Ferrara (authorization number: 160295). Written informed consent was obtained for all individual participants involved in the study and was signed by a parent.

## Procedure

Neonates for whom consent was obtained from parents were observed two times in an open-bay setting with four cots, respectively before and after scheduled feedings, while they were lying supine. All of the video-recordings took place in the afternoon, while the neonates were not receiving any stimulation through routine nursing or medical care, and lasted around 30 minutes (M = 31.47, SD = 10.48). All of the neonates were video-recorded both before and after feeding, except for the members of one twin pair who were observed only before feeding. Hence, analyses were conducted on 114 video-recordings. The observations took place at the Neonatal Intensive Care Unit (NICU) of the S. Anna University Hospital of Ferrara (Italy). In order to enhance replicability, the study protocol is available on protocols.io at https://dx.doi.org/10.17504/protocols.io.b337qqrn.

## Behavioral coding

Frame by frame behavioral analysis of video-recordings was performed by two independent coders expert in behavioral micro-analysis (with the secondary coder examining 30% of the videorecordings, n = 36), using ELAN, a professional software for the creation and management of *complex annotations* on video and audio (Max Planck Institute for Psycholinguistics, The Language Archive, Nijmegen, The Netherlands; http://tla.mpi.nl/tools/tla-tools/elan/).

**Yawn coding.** Yawns were identified holistically according to the *System for Coding Perinatal Behavior* (*SCPB*) [50], which is based in part on selected facial Action Units (AUs) from the anatomically based *Baby FACS*: *Facial Action Coding System for Infants and Young Children* [51] and previous studies in the literature [18, 52]. For reliability assessment purposes, once an event was identified as a yawn based on this description, the onset and offset were scored respectively at the first and last frame where mouth opening was visible. The SCPB was used in recent studies to code yawns and other behaviors in fetuses [29] and preterm neonates [53].

Yawning is defined in the SCPB as a stereotyped behavior characterized by a slow mouth opening with deep inspiration, followed by a brief apnea and a short expiration and mouth closing, typically accompanied by limb stretching [52]. The expansion of the pharynx can quadruple its diameter, while the larynx opens up with maximal abduction of the vocal cords [18]. One of the characteristic features of yawning [51] is its timing, consisting in a progressive acceleration, followed by an abrupt deceleration in the intensity of the facial muscle Action Units (AUs) involved, designated by numeric codes and verbal labels. Yawning usually emerges from a relaxed face, initially involving mouth opening (AUs 25, 26, 27) and eyes closing (AU 43E), followed by upper eyelid drooping (AU 43A-D), flattened tongue on the bottom of the mouth (AU 75) and usually swallowing (AU 80). During the plateau brow knitting (AU 3), brow knotting (AU 4), nose wrinkling (AU 9), lateral lip stretching (AU 20), nostril dilatation (AU 38) and head tilting back (AU 53) also typically occur.

**Quiet sleep coding.** Quiet sleep (regular sleep or non-REM sleep, QS) periods were scored when the following conditions were met for at least three consecutive minutes: primarily abdominal respiration which is regular in rhythm and constant in amplitude, fully closed eyelids and no movements except for occasional startles, sudden jerks or rhythmic mouthing [54–56].

### Data analysis

Inter-rater reliability between the primary and secondary coder was calculated using Cohen's Kappa, with a satisfactory level of agreement for all the variables coded. In particular, reliability was assessed for the occurrence of yawning by adopting a one-second threshold both for onset and offset (Kappa = 1), while a 30 seconds threshold was used for calculating Kappa for quiet sleep scores (Kappa = 0.82).

Multilevel Poisson random intercept regressions were adopted in order to account for the hierarchical structure of the data set, with observations (before or after feeding) nested in subjects, nested in twin pairs, nested in classes (singleton or twin). The offset variable was the natural logarithm of the observation time.

Four random intercept Poisson regressions were conducted on the entire sample to compare yawning frequencies before and after feeding, and to control for the effects of QS-ratio (defined as the fraction of observation time spent in QS), gender, and class (twin or singleton). Moreover, a four-level random intercept regression was fitted on the twins sub-sample in order to assess the effect of birth order, and linear Poisson regressions were used to evaluate the potential effects of age variables (GA, CH and PMA) on the yawning frequencies observed in the two conditions. In fact, the developmental trajectories of preterm neonates can only be described in terms of the joint analysis of these three age variables [49, 57]. Therefore, although no specific hypothesis was formulated regarding potential associations between age variables and yawning rates, such relationships were investigated. Moreover, a multilevel linear regression was used to test whether mean yawn durations were different before and after feeding.

In a further effort to discriminate hunger and sleep-related yawn modulations, once we confirmed that no yawn was scored during QS, these periods were excluded from analysis. This was accomplished by calculating, for each observation, the portion of observation time spent in behavioral states other than QS, subtracting the time spent in QS from the observation time. The resulting time (non-QS) was therefore used as logarithmic offset for a four-level Poisson regression including condition (before or after feeding), gender and class (twin or singleton) as predictors. In order to ensure a higher homogeneity of conditions and to eliminate the possible noise due to the inclusion of observations whereonly a short time is spent outside of quiet sleep, the video recordings where the remaining observation time after excluding quiet sleep (non-QS) was shorter than the arbitrary threshold of 1,000 s (16 minutes and 40 seconds) were excluded from this analysis.

Bonferroni significance was set at .00625 to account for multiple testing.

All analyses were carried out in the R statistical environment, version 4.0.2 [58], using the lmerTest package [59].

## Results

### Descriptive statistics

Overall, 35 out of 58 neonates (60%) yawned at least once across the two observations (before and after feeding). Before feeding, the average rate of yawns per hour was 3.60 (SD = 5.37), while after feeding it was 1.50 (SD = 3.32).

### Condition-related modulation

The hierarchical regression fitted to the entire sample revealed a significant effect of the condition, $F(5,109) = 0.50$, $\beta = 0.98$, $p < 0.001$, demonstrating higher yawning frequencies in the observation before feeding compared with after feeding, as shown in Fig 1. A second hierarchical model, including condition and QS-ratio (fraction of observation time spent in QS) as predictors, revealed lower yawning frequencies associated with higher QS-ratios, $F(6,108) = 0.64$,

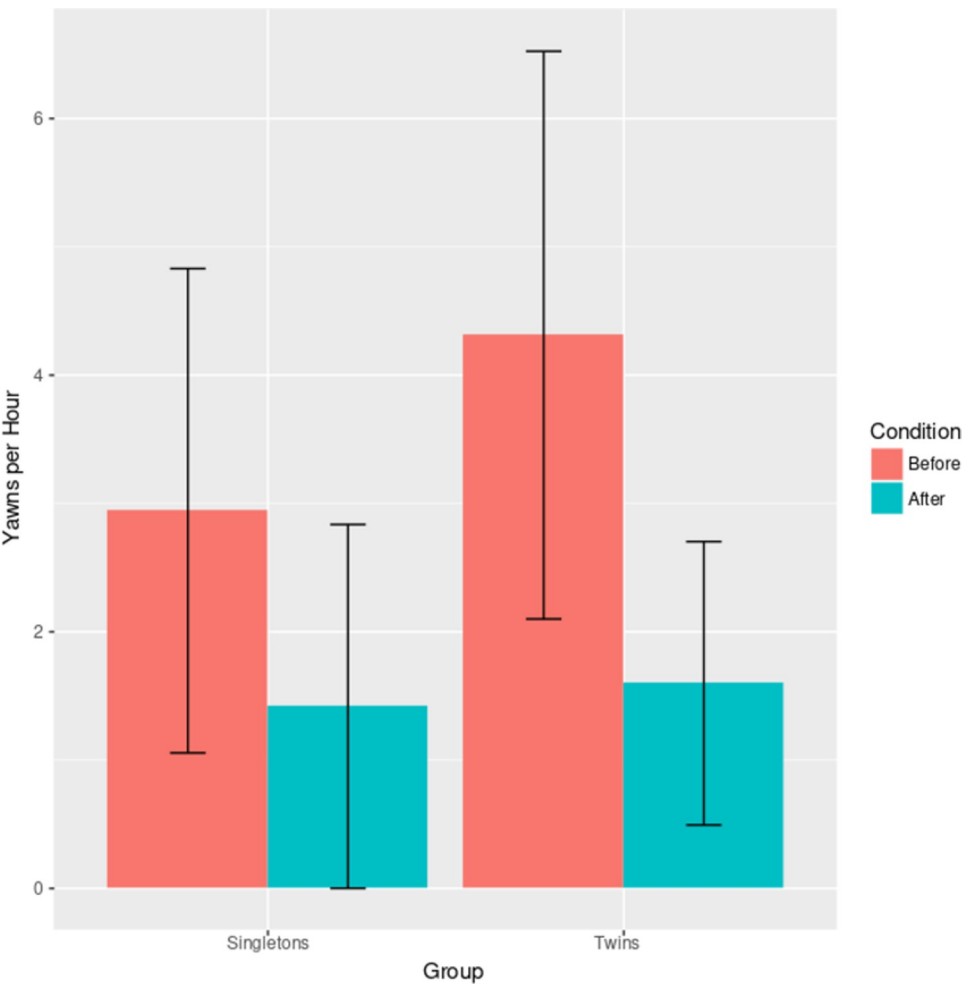

**Fig 1. Yawning frequencies before and after feeding.** Error bars represent 95% confidence intervals.

$\beta$ = -3.57, p < 0.001, and confirmed the increase in yawn frequencies before feeding F(6,108) = 0.64, $\beta$ = 0.75, p < 0.001. Two additional regressions that included gender F(7,107) = 0.63, $\beta$ = 0.28, p = 0.47 and class of subjects (twins vs singletons) F(7,107) = 0.65, $\beta$ = 0.65, p = 0.25) as well as condition and QS-ratio as predictors, did not show significant differences, nor improve the model fit. Mean yawn durations were not affected by condition (before vs. after feeding), F(1,30) = 1.03, $\beta$ = 0.31, p = .318.

### Birth order

A three-level random intercept Poisson regression conducted on twins only (n = 28), while confirming the effect of condition (before vs. after feeding), F(5, 49), = 0.58, $\beta$ = 1.07, p < 0.001, revealed a significant effect of birth order in twins, F(5,49) = 0.58, $\beta$ = 1.04, p < 0.001, with the second born showing increased overall yawning frequencies compared with the first born.

### Age variables

Age-based comparisons did not find effects of PMA, GA or CH on yawning frequencies before or after feeding.

## Hunger-related modulation

After verifying that no yawn had occurred during QS, the non-QS offset was calculated, and observations where the remaining time (non-QS) was less than 1,000 s were excluded, leaving 80 observations (43 before and 37 after feeding) across 49 neonates (23 twins and 26 singletons). The regression fitted on this sub-samples using non-QS as logarithmic offset confirmed the effect of hunger, $F_{(6,74)} = 0.55$, $\beta = 0.78$, $p < 0.001$, in modulating yawn frequencies, even after checking for effects of gender $F_{(6,74)} = 0.55$, $\beta = 0.07$, $p = 0.873$, and class (twins vs singletons), $F_{(6,74)} = 0.55$, $\beta = 0.45$, $p = 0.290$.

## Discussion

Both singletons and twins displayed higher yawning frequencies before feeding compared with after feeding, confirming the hypothesis of a condition-related modulation of yawning. Because the effect of condition was still significant after checking for the effect of the portion of observation time spent in quiet sleep, and even after excluding quiet-sleep periods from the analysis, we can conclude that this modulation is not entirely explained by a difference in the distribution of behavioral states but is at least in part directly due to hunger. This finding is in line with the existing literature on hunger-related modulation of yawning frequencies in adults [4, 5] and shows that this mechanism is already observable in preterm neonates. Moreover, this form of modulation is consistent with the brain cooling hypothesis, as previous research has shown that feeding in preterm neonates is integrated into a heat production episode [39–40]. Moreover, our results confirmed the absence of yawning during quiet sleep in preterm neonates [28, 30], showing the importance of considering quiet sleep for behavioral research in early infancy.

Finally, the increased yawning frequencies found in the second born twin, known to be prone to higher neonatal morbidity [47, 60], is consistent with the hypothesis that yawning behavior might be affected by perinatal clinical conditions [35, 61]. This seems to confirm that analysis of yawning is a promising tool for neurobehavioral assessment, potentially allowing clinicians and researchers to identify at-risk infants through early observation, both during the fetal and neonatal periods [62, 63]. Overall, our results are consistent with the hypothesis that yawning frequencies in preterm neonates are modulated by two separate cholinergic-related factors—respectively, hunger and sleep-related factors—as well as by a possibly ACTH-mediated stress-related condition (i.e. being a second born twin). These findings, although preliminary and based only on behavioral data, might indicate that several distinct neuropharmacological pathways that have been found to be involved in yawn modulation [64] are already observable in preterm neonates. Furthermore, demographic variables, including gender and age measures (GA, CH and PMA) were not found to affect yawning frequencies, suggesting that the observed pattern can be generalized to healthy preterm neonates from at least 32 weeks PMA. These results represent an advance in efforts to tease apart the effects of between- as well as within-subject factors that may influence the frequency of yawning in preterm neonates. The findings are consistent with the hypothesis that yawning in preterm neonates is modulated by partially autonomous cholinergic and ACTH-mediated processes.

Future research should address some limitations of the present study, by investigating the specific effect of different behavioral states, including both quiet and active sleep, as well as the potential associations between yawning frequencies and state transitions or instability. Moreover, additional studies will be needed in order to directly test the hypothesis that stress and hunger-related modulations in neonates are in fact ascribable to cholinergic and ACTH-mediated pathways, as postulated by Collins and Eguibar [64]. In particular, further research is needed to confirm whether the effect of birth order on yawning rates in twins is due to stress-

related factors associated with birth or to other variables and whether hunger or stress-related modulation of yawning can be explained in terms of brain thermoregulation.

Finally, although the exploratory analysis of yawn durations before and after feeding did not show any difference, other behavioral studies might be useful to test whether yawns associated with different conditions and modulatory mechanisms also present some particularities in terms of morphology, intensity or temporal dynamics, as recently shown for some species of apes [22, 24].

## Supporting information

**S1 File. Dataset.**
(CSV)

**S2 File. R code for data analysis.**
(R)

## Acknowledgments

We are deeply indebted to the nursing staff at the Neonatal Intensive Care Unit (NICU) of the S. Anna University Hospital of Ferrara, and in particular to Maria Grazia Cristofori, for their invaluable cooperation. We also wish to thank Tiziana Aureli for commenting an earlier version of the study.

## Author Contributions

**Conceptualization:** Roberta Panebianco, Marco Dondi.

**Data curation:** Damiano Menin.

**Formal analysis:** Damiano Menin, Roberta Panebianco.

**Investigation:** Damiano Menin, Elisa Ballardini, Roberta Panebianco, Giampaolo Garani, Caterina Borgna-Pignatti, Marco Dondi.

**Methodology:** Damiano Menin, Roberta Panebianco, Giampaolo Garani, Caterina Borgna-Pignatti, Harriet Oster, Marco Dondi.

**Project administration:** Elisa Ballardini, Giampaolo Garani, Caterina Borgna-Pignatti, Marco Dondi.

**Resources:** Caterina Borgna-Pignatti, Marco Dondi.

**Supervision:** Elisa Ballardini, Giampaolo Garani, Harriet Oster, Marco Dondi.

**Visualization:** Damiano Menin.

**Writing – original draft:** Damiano Menin, Marco Dondi.

**Writing – review & editing:** Damiano Menin, Elisa Ballardini, Roberta Panebianco, Giampaolo Garani, Caterina Borgna-Pignatti, Harriet Oster, Marco Dondi.

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
