## [Decision Letter · Decision Letter 0]

16 Feb 2022

PONE-D-22-01955Factors affecting yawning frequencies in preterm neonatesPLOS ONE

Dear Dr. Dondi,

Thank you for submitting your manuscript to PLOS ONE. After careful consideration, I feel that it has merit but does not fully meet PLOS ONE’s publication criteria as it currently stands. Therefore, I invite you to submit a revised version of the manuscript that addresses the points raised during the review process.

I carefully reviewed the manuscript and have now received reviews from two additional experts on yawning. Both referees indicate an interest in the study, but each also presents some concerns. Overall, there have been a number of issues raised with regards to the statistics, methods, and conclusions that currently prevent publication in the journal. Therefore, major revisions are recommended. Reviewer #1 calls for a new analysis of the data, and Reviewer #2 also questions the analytic strategy. I have also provided some comments about this below. In addition, both reviewers bring up concerns about the reported details of the observation period and other methodological considerations that should be addressed. I have also provided some comments about potential confounds in the study that should be addressed. Both reviewers also specify areas where some conclusions may not be justified based on the data presented. I also provide some suggestions for incorporating past research on feeding and thermoregulation in infants. In addition, both reviewers also bring up concerns about the identification of hunger (i.e., First, how can hunger be ascertained, particularly given that the feedings were scheduled rather than on-demand? Second, why hunger would be a trigger for yawning?). 

We look forward to receiving your revised manuscript.

Kind regards,

Andrew C Gallup, Ph.D.

Academic Editor

PLOS ONE

Journal Requirements:

When submitting your revision, we need you to address these additional requirements. 1. Please ensure that your manuscript meets PLOS ONE's style requirements, including those for file naming. The PLOS ONE style templates can be found at https://journals.plos.org/plosone/s/file?id=wjVg/PLOSOne_formatting_sample_main_body.pdf and https://journals.plos.org/plosone/s/file?id=ba62/PLOSOne_formatting_sample_title_authors_affiliations.pdf 2. Please include a copy of Table 1 which you refer to in your text on page 3.

Additional Editor Comments:

It could be acknowledged how the increase in yawning prior to feeding is consistent with a role in thermoregulation. The authors cite some research linking yawning to thermoregulation, but not with respect to the current findings. Previous studies have shown that feeding in newborn infants is integrated into a heat production episode, i.e., thermoregulatory feeding hypothesis (Himms-Hagen, 1995 Obesity Research). This is particularly evident for on-demand feedings, which are preceded by a rapid rise in body temperature (Himms-Hagen, 1997 Obesity Research; Chardon et al., 2006 Obesity). However, scheduled feedings in newborns are also followed by a progressive decrease in core temperature (Chardon et al., 2006 Obesity). Thus, the reduced yawning post-feeding in the current study is consistent with a thermoregulatory function to yawning. Prior research has shown that yawns are triggered during rises in brain/body temperature (e.g., Shoup-Knox et al., 2010 Frontiers in Neuroscience) and that the frequency of yawning increases during rising ambient temperature and diminishes at low ambient temperatures (e.g., Massen et al., 2014 Physiology & Behavior). Similarly, warming of the carotid arteries increases yawning while cooling of this blood flow to the brain decreases yawning (e.g., Ramirez et al., 2019 Physiology & Behavior). The thermoregulatory events surrounding feeding map on to the general pattern of increased yawning when we are warm and decreased yawning when we are cold or at thermal homeostasis.

Additional comments:

It appears that a large number of statistical tests were performed, so the authors should include corrections for multiple tests.

Circadian rhythm is a confound to this study since recordings before and after feedings always vary in timing, whereby the former always precedes the later. This is a potentially important concern given the circadian variation in yawning.

Reviewers' comments:

Reviewer's Responses to Questions

**Comments to the Author**

1. Is the manuscript technically sound, and do the data support the conclusions?

Reviewer #1: No

Reviewer #2: Yes

2. Has the statistical analysis been performed appropriately and rigorously? 

Reviewer #1: Yes

Reviewer #2: Yes

3. Have the authors made all data underlying the findings in their manuscript fully available?

Reviewer #1: Yes

Reviewer #2: Yes

4. Is the manuscript presented in an intelligible fashion and written in standard English?

Reviewer #1: Yes

Reviewer #2: Yes

5. Review Comments to the Author

Reviewer #1: The study aims at exploring the potential factors influencing yawning frequencies in hospitalized preterm neonates to try to differentiate the hunger and sleep-related effects on the yawning phenomenon. The authors found that the neonates yawned more frequently before than after being fed feeding and that this increase was independent from the time spent in quiet sleep by the subjects. By studying twins, the authors also found that the second born yawned more that the first born. This result was discussed in the light of the higher risk of morbidity and mortality of the second born compared to the first born twins. So the authors conclude that yawning activity can be modulated by hanger, vigilance and stress and that all these factors act as separate mechanism. Although the topic is extremely interesting, I think the study suffers some methodological weaknesses that, I really hope, can be solved by an accurate revision and a new analysis of the data.

Line 115 – “All of the video-recordings took place in the afternoon” why only in the afternoon. Since a daily fluctuation is reported in yawning distribution, why limiting the data collection to a restricted part of the day?

Lines 115-117 – “…, while the neonates were not receiving any stimulation through routine nursing or medical care”. I think the potential stimulation is an important point. For this reason, additional information on what the neonates could potentially perceive is extremely important. Were the neonates isolated from the other ones? Could they hear other neonates crying, for example? Hearing others crying can increase the level of anxiety in the subjects. If before feeding, the number of crying neonates increased and they could hear each other, it is possible that the real cause at the basis of the increase of yawning is not hunger, but simply an increase of arousal. How did the authors measure and ascertain that the neonates were actually hungry during the time block “before feeding”. We can suppose they were, but a criterion of evaluation needs to be clearly used and explained to make the experiment replicable.

Lines 118-119 – “and lasted around 30 minutes (M =31.47, SD =10.48). All of the neonates were video recorded both before and after feeding, except for the members of one twin pair who were observed only before feeding.” How many minutes of videos before and how many after feeding? Were the time windows balanced? And if not, did the authors calculated the exact number of yawns per minute of observing time to have a reliable estimation of yawning frequency? This is a fundamental piece of information and it should be given in the text.

Line 142 - better inhalation and exhalation than “inspiration” and “expiration”

Line 142 – To operationally define the yawning pattern, the authors should limit the description to what can be visually inspected and objectively reported. So, in the paragraph in which they describe the criteria used to record each yawning event it should be better avoid citing “The expansion of the pharynx can quadruple its diameter, while the larynx opens up with maximal abduction of the vocal cords” unless these behaviors were actually scored during the data collection.

Line 165 – “Because durations were not scored for yawns…” why did the authors not score yawn durations? From the literature reported in the introduction, duration seems to be a good factor at the basis of yawning variability.

Line 186-189 – “In order to eliminate the possible noise due to the inclusion of observations where most of the time was spent in quiet sleep, the video recordings where the remaining observation time after excluding quiet sleep (non-QS) was shorter than the arbitrary threshold of 1,000 s (16 minutes and 40 seconds) were excluded from this analysis.” Which is the rationale at the basis of this choice? Since I am sure that there is a reason, I think the authors should clearly explain it.

Line 244-251 – The authors explain the higher frequency of yawning in the second born twins in the light of their higher risk of morbidity and mortality compared to the first born twins. So the authors conclude that yawning activity can be modulated by stress. However, data on the perinatal clinical conditions of babies are missing. In absence of any data on stress experienced by the neonates, nothing can be said about the effect of stress on yawning frequency variability.

I do not think that the conclusions are supported by the data, however since the authors have the videos and (I guess) also the clinical data of the neonates, they could use these supporting information to re-analyze the data on yawning.

Minor points

Introduction

Line 63 – eliminate “of”

Reviewer #2: Dear Authors,

you report here on a study examining factors that affect/modulate yawn frequencies in preterm neonates. Overall I think the study is well designed and executed, and the results are clear and comprehensible. As a results, some of your conclusions are therefore also justified. However, it remains a bit unclear what the actual rationale for this study was; i.e. why is it specifically interesting to study yawning in preterm neonates? What are the actual hypotheses. You summarize some of the existing hypotheses about yawning, but seem to convolute proximate and ultimate accounts and as a consequence present them as being mutually exclusive, which they in fact do not need to be. Nevertheless, the pletoria of hypotheses might be a reason to investigate further, yet the rationale to then look at pre-term neonates remains obscured, and without proper predictions at the end of the introduction the study does feel a bit like a fishing expedition. Therefore, I would like to see the study better embedded in the theoretical framework, creating a more balanced and useful account of your findings. Apart from that, I only have some minor suggestions/comments:

l. 41: Please refer to Massen et al. 2021 Communications Biology, as this study is a much more elaborate account of the ubiquity of yawning in vertebrates (more species, more recent)

l. 49: If you want to refer to Apes, then please also cite van Berlo et al. 2020 Scientific Reports, which shows contagious yawning in Orangutans, since the current references only refer to homo sapiens and the two pan species

l. 55: Not all of the hypothese above are functional ones, some are mechanistic and therefore, not all of these hypothese are mutually exclusive. Please make sure you have the framework right here, as it allows for a better understanding of where to place your results in the end.

l. 67: Please cite Gallup et al. 2021 Scientific Reports that also shows the effect of amount of sleep on yawn modulation

l. 69-72: What is the hypothesis? Why should hunger lead to a modulation of yawning?

l. 91: What are stress-related yawns? Please provide a reference, also to inform us about the hypothesis here, which is kind of missing.

l. 101 what does appropriate for gestational age mean? This is probably very familiar lingo for physicians, but you are reporting to a general journal, so please take into account the various backgrounds of your readers.

l. 114-122: The number of observation minutes seems rather low to me. Do you have any comparisons that allow us to gather how well such observational scheme represents the complete picture? and how prone it now is to random error?

l.170. why did you use the natural log. of time for the offset and not just the actual time?

l.179. If there are no specific hypotheses, why would you reduce the power of your analyses (on a relatively small sample) by adding additional variables?

l.186. Similarly, why was gender included? Are there any specific hypotheses about gender effects?

l.2019-221. Please remove this sentence as it is redundant. There were no effects. p = 016 is not CLOSE to significance.

l.246ff This conclusion is not justified as the relationship is indirect.

6. PLOS authors have the option to publish the peer review history of their article (what does this mean?). If published, this will include your full peer review and any attached files.

Reviewer #1: No

Reviewer #2: No

It could be acknowledged how the increase in yawning prior to feeding is consistent with a role in thermoregulation. The authors cite some research linking yawning to thermoregulation, but not with respect to the current findings. Across different species, including humans, yawns are triggered during rises in brain/body temperature and temperatures decrease following this behavior (Shoup-Knox et al., 2010; Eguibar et al., 2017; Gallup & Gallup, 2010; Gallup et al., 2017). The frequency of yawning can be reliably altered by ambient temperature changes (Massen et al., 2014; Eldakar et al., 2015), whereby yawning increases during rising temperature and diminishes at low temperatures. Similarly, manipulations to brain/body temperature are yawning in predicted ways (Gallup & Gallup, 2007; Ramirez et al., 2019). In particular, warming of the neck and skull are associated with increased yawning, while cooling of these surfaces diminishes yawning. Together, rises in temperature are linked with increases in yawning, while decreases in temperature are linked with reduced yawning.

This literature pertains to the current study because, in newborn infants, feeding and thermoregulation are also connected. Previous studies have shown that feeding in newborn infants is integrated into a heat production episode, i.e., thermoregulatory feeding hypothesis (Himms-Hagen, 1995). This is particularly evident for on-demand feedings, which are preceded by a rapid rise in body temperature (Himms-Hagen, 1997; Chardon et al., 2006). However, scheduled feedings in newborns, which were observed in the current study, are also followed by a progressive decrease in core temperature (Chardon et al., 2006). Thus, the reduced yawning post-feeding in the current study is consistent with a thermoregulatory cooling function to yawning. 

Both reviewers bring up concerns about the identification of hunger. First, how can hunger be ascertained, particularly given that the feedings were scheduled rather than on-demand? Second, why hunger would be a trigger for yawning?

Circadian rhythm is a confound, before and after feedings always vary in timing, whereby the former always preceded the later.

Corrections for multiple tests

---

## [Author Response · Author response to Decision Letter 0]

1 Apr 2022

PONE-D-22-01955

Factors affecting yawning frequencies in preterm neonates

PLOS ONE

Dear Dr. Dondi,

Thank you for submitting your manuscript to PLOS ONE. After careful consideration, I feel that it has merit but does not fully meet PLOS ONE’s publication criteria as it currently stands. Therefore, I invite you to submit a revised version of the manuscript that addresses the points raised during the review process.

I carefully reviewed the manuscript and have now received reviews from two additional experts on yawning. Both referees indicate an interest in the study, but each also presents some concerns. Overall, there have been a number of issues raised with regards to the statistics, methods, and conclusions that currently prevent publication in the journal. Therefore, major revisions are recommended. Reviewer #1 calls for a new analysis of the data, and Reviewer #2 also questions the analytic strategy. I have also provided some comments about this below. In addition, both reviewers bring up concerns about the reported details of the observation period and other methodological considerations that should be addressed. I have also provided some comments about potential confounds in the study that should be addressed. Both reviewers also specify areas where some conclusions may not be justified based on the data presented. I also provide some suggestions for incorporating past research on feeding and thermoregulation in infants. In addition, both reviewers also bring up concerns about the identification of hunger (i.e., First, how can hunger be ascertained, particularly given that the feedings were scheduled rather than on-demand? Second, why hunger would be a trigger for yawning?). 

We look forward to receiving your revised manuscript.

Kind regards,

Andrew C Gallup, Ph.D.

Academic Editor

PLOS ONE

Journal Requirements:

2. Please include a copy of Table 1 which you refer to in your text on page 3. 

Additional Editor Comments:

Q1: It could be acknowledged how the increase in yawning prior to feeding is consistent with a role in thermoregulation. The authors cite some research linking yawning to thermoregulation, but not with respect to the current findings. Previous studies have shown that feeding in newborn infants is integrated into a heat production episode, i.e., thermoregulatory feeding hypothesis (Himms-Hagen, 1995 Obesity Research). This is particularly evident for on-demand feedings, which are preceded by a rapid rise in body temperature (Himms-Hagen, 1997 Obesity Research; Chardon et al., 2006 Obesity). However, scheduled feedings in newborns are also followed by a progressive decrease in core temperature (Chardon et al., 2006 Obesity). Thus, the reduced yawning post-feeding in the current study is consistent with a thermoregulatory function to yawning. Prior research has shown that yawns are triggered during rises in brain/body temperature (e.g., Shoup-Knox et al., 2010 Frontiers in Neuroscience) and that the frequency of yawning increases during rising ambient temperature and diminishes at low ambient temperatures (e.g., Massen et al., 2014 Physiology & Behavior). Similarly, warming of the carotid arteries increases yawning while cooling of this blood flow to the brain decreases yawning (e.g., Ramirez et al., 2019 Physiology & Behavior). The thermoregulatory events surrounding feeding map on to the general pattern of increased yawning when we are warm and decreased yawning when we are cold or at thermal homeostasis.

RESPONSE: We thank the Editor for suggesting this convincing interpretation. We introduced your insightful comments in our manuscript, by mentioning this hypothesis in the introduction (lines 58-64) as well as in the discussion (lines 305-308).

Additional comments:

Q2: It appears that a large number of statistical tests were performed, so the authors should include corrections for multiple tests.

RESPONSE: . In the new version of the manuscript, we adopted Bonferroni correction (line 190, see below). This very conservative correction, however, does not imply any further change, as all significant associations/effects have p<.001. This is a valid suggestion, although somehow controversial (see, e.g., Gelman et al., 2012; https://doi.org/10.1080/19345747.2011.618213).

"Bonferroni significance was set at .00625 to account for multiple testing."

Q3: Circadian rhythm is a confound to this study since recordings before and after feedings always vary in timing, whereby the former always precedes the later. This is a potentially important concern given the circadian variation in yawning.

RESPONSE: This could be a concern if our study focused on older children or adults, but preterm neonates in particular don't show circadian rhythms before one month corrected age (see e.g. Ivars et al., 2017, PLoS ONE, https://doi.org/10.1371/journal.pone.0182685). Also, Giganti et al., 2007 (https://doi.org/10.1016/j.infbeh.2007.03.005) investigated potential circadian variations in yawning frequencies of preterm neonates and found none. Moreover, we have to point out that our methodological choices were limited due to the challenges of performing research in a Neonatal Intensive Care Unit (NICU).

Reviewers' comments:

Reviewer's Responses to Questions

Comments to the Author

1. Is the manuscript technically sound, and do the data support the conclusions?

Reviewer #1: No

Reviewer #2: Yes

2. Has the statistical analysis been performed appropriately and rigorously?

Reviewer #1: Yes

Reviewer #2: Yes

3. Have the authors made all data underlying the findings in their manuscript fully available?

Reviewer #1: Yes

Reviewer #2: Yes

4. Is the manuscript presented in an intelligible fashion and written in standard English?

Reviewer #1: Yes

Reviewer #2: Yes

5. Review Comments to the Author

Reviewer #1: The study aims at exploring the potential factors influencing yawning frequencies in hospitalized preterm neonates to try to differentiate the hunger and sleep-related effects on the yawning phenomenon. The authors found that the neonates yawned more frequently before than after being fed feeding and that this increase was independent from the time spent in quiet sleep by the subjects. By studying twins, the authors also found that the second born yawned more that the first born. This result was discussed in the light of the higher risk of morbidity and mortality of the second born compared to the first born twins. So the authors conclude that yawning activity can be modulated by hanger, vigilance and stress and that all these factors act as separate mechanism. Although the topic is extremely interesting, I think the study suffers some methodological weaknesses that, I really hope, can be solved by an accurate revision and a new analysis of the data.

Q1: Line 115 – “All of the video-recordings took place in the afternoon” why only in the afternoon. Since a daily fluctuation is reported in yawning distribution, why limiting the data collection to a restricted part of the day?

RESPONSE: The choice to limit data collection to the afternoons was due to different reasons: 

The policies and schedules of the Neonatal Intensive Care Unit (NICU) don't allow great flexibility in data collection, as neonates that are hospitalized in this context are extremely fragile.

Considering the difficulties that are always associated with the effort to recruit an acceptable sample of preterm neonates, we deemed best to keep the conditions of observations as homogeneous as possible. In fact, if our sample was heterogeneous with regard to potential confounders, we probably would not have enough statistical power to test our hypotheses.

Finally, we have to note that, although diurnal variations of yawning frequencies are well known in adults, Giganti, Hayes, Cioni & Salzarulo (2007) have shown no such variations in preterm neonates, likely because of the immaturity of circadian and homeostatic control of sleep and wake.

Q2: Lines 115-117 – “…, while the neonates were not receiving any stimulation through routine nursing or medical care”. I think the potential stimulation is an important point. For this reason, additional information on what the neonates could potentially perceive is extremely important. Were the neonates isolated from the other ones? Could they hear other neonates crying, for example? Hearing others crying can increase the level of anxiety in the subjects. If before feeding, the number of crying neonates increased and they could hear each other, it is possible that the real cause at the basis of the increase of yawning is not hunger, but simply an increase of arousal. How did the authors measure and ascertain that the neonates were actually hungry during the time block “before feeding”. We can suppose they were, but a criterion of evaluation needs to be clearly used and explained to make the experiment replicable.

RESPONSE: Neonates were not isolated, they were in an open-bay setting as their conditions were not critical (this information was reported in the Participants section, lines 114, 115: "Neonates for whom consent was obtained from parents were observed two times in an open-bay setting with four cots, respectively before and after scheduled feedings, while they were lying supine"). However, because of their immaturity, cries are very rare in preterm neonates before term-equivalent age, and when they happen they are nowhere near as loud as in full-term neonates and infants (see, e.g., Ranger, Johnston, & Anand, 2007; Seminars in Perinatology). During the videorecordings that were included in this study, in particular, no episode of audible crying was recorded either by the observed neonate nor by other neonates in the room.

Regarding the latter issue, we could argue that the fact that feedings were scheduled actually improves replicability, as this approach reduces the arbitrary variability that might be associated with responsive feeding. Although we don't have a definite way of ascertain whether and how much all neonates were hungry, this is true in general for pre-verbal subjects, and in particular for preterm neonates, who are often fed on schedule exactly because of their inability to self-regulate feeding (McCain, 2003, Neonatal Network). However, both the modulation in quiet sleep distributions and the fact that feeding was successful in all cases and neonates were able to coordinate sucking and swallowing indicate that neonates were actually hungry. The conditions "before" and "after feeding", moreover, were objectively different as in one case neonates had just been fed while in the other case they did not receive food during the previous three hours.

Q3: Lines 118-119 – “and lasted around 30 minutes (M =31.47, SD =10.48). All of the neonates were video recorded both before and after feeding, except for the members of one twin pair who were observed only before feeding.” How many minutes of videos before and how many after feeding? Were the time windows balanced? And if not, did the authors calculated the exact number of yawns per minute of observing time to have a reliable estimation of yawning frequency? This is a fundamental piece of information and it should be given in the text.

RESPONSE: This concern is about one of the very basic methodologies of developmental and behavioral psychology. As the reviewer pointed out, one common approach to this issue is to use rates per minute (which, in order to improve clarity are now reported in the "descriptive statistics" subsection of results). However, because, in accordance with the count nature of our data and of a preliminary analysis of its distribution we adopted Poisson regressions as analytic method, instead of calculating the rate of yawns per minute, we specified in our models the log-transformed value of the observation time as the offset. This is, to the best of our knowledge, the state-of-the-art approach when dealing with count data with varying exposures (see, e.g., Hutchinson & Holtman, 2005: https://doi.org/10.1002/nur.20093). In general, although it is somehow common practice to treat count data as interval/ratio variables, Poisson methods are preferable when dealing with skewed count data (Nussbaum et al., 2008: https://doi.org/10.4135/9781412995627.d26).

Q4: Line 142 - better inhalation and exhalation than “inspiration” and “expiration”

Line 142 – To operationally define the yawning pattern, the authors should limit the description to what can be visually inspected and objectively reported. So, in the paragraph in which they describe the criteria used to record each yawning event it should be better avoid citing “The expansion of the pharynx can quadruple its diameter, while the larynx opens up with maximal abduction of the vocal cords” unless these behaviors were actually scored during the data collection.

RESPONSE: Both these comments refer to a description extrapolated from the System for Perinatal and Infant Coding that was already used in previous studies (see references #27 and #43 as well as Menin, Aureli & Dondi, 2022, PLOS ONE) and is based on FACS (Ekman & Friesen, 1978) and Baby FACS (Oster, 2015). As for the choice between inspiration/expiration and inhalation/exhalation, we did not find evidence of any preference for the latter either in common use or in the scientific literature. Moreover, although not all the behavioral components of yawning can be always (or often) observed when coding neonatal behavior, we argue that having a description as detailed and comprehensive as possible can only increase both the validity and reliability of yawn identification, an aspect that is often neglected in yawning research (see e.g. Menin et al., 2019, PLOS ONE).

Q5: Line 165 – “Because durations were not scored for yawns…” why did the authors not score yawn durations? From the literature reported in the introduction, duration seems to be a good factor at the basis of yawning variability.

RESPONSE: We are aware of the promising evidence that points at yawn duration as a potential marker of different types of yawns in some non-human primates (Deputte, 1994; Leone et al., 2014; Zannella et al., 2020) and as a predictor of brain weight (Gallup et al., 2016) and we find this line of research extremely interesting. On the other hand, to the best of our knowledge no study has found similar differences in humans. More importantly, based on the literature we analyzed, we had no hypothesis about potential differences in yawn durations, either between conditions and between different groups of subjects. However, we agree with the reviewer that this study might present an opportunity for an exploratory investigation of potential variations in yawn duration across different conditions and between different groups. Therefore, we coded yawn durations and analyzed the potential association between average yawn duration and condition (before vs after feeding) using a linear regression, and revised accordingly all sections of the manuscript (introduction, methods, results and discussion).

Q6: Line 186-189 – “In order to eliminate the possible noise due to the inclusion of observations where most of the time was spent in quiet sleep, the video recordings where the remaining observation time after excluding quiet sleep (non-QS) was shorter than the arbitrary threshold of 1,000 s (16 minutes and 40 seconds) were excluded from this analysis.” Which is the rationale at the basis of this choice? Since I am sure that there is a reason, I think the authors should clearly explain it.

RESPONSE: We changed this sentence in an effort to make it more clear, although we think that the response to the reviewer's Q3 should also clarify this point, as non-QS time was used as log-transformed offset (exposure time) in this model. We think that this approach is better suited to ensure the robustness of these results, but we also have to point out that this regression gives very similar results when the model is fitted on the whole dataset (without excluding those with non-QS observation time < 1000s).

"In order to ensure a higher homogeneity of conditions and to eliminate the possible noise due to the inclusion of observations where only a short time is spent outside of quiet sleep, the video recordings where the remaining observation time after excluding quiet sleep (non-QS) was shorter than the arbitrary threshold of 1,000 s (16 minutes and 40 seconds) were excluded from this analysis."

Q7: Line 244-251 – The authors explain the higher frequency of yawning in the second born twins in the light of their higher risk of morbidity and mortality compared to the first born twins. So the authors conclude that yawning activity can be modulated by stress. However, data on the perinatal clinical conditions of babies are missing. In absence of any data on stress experienced by the neonates, nothing can be said about the effect of stress on yawning frequency variability.

I do not think that the conclusions are supported by the data, however since the authors have the videos and (I guess) also the clinical data of the neonates, they could use these supporting information to re-analyze the data on yawning.

RESPONSE: We have access to the videos and the medical records of the participants, but, in particular because we excluded from data collection neonates with serious pathologies (congenital anomalies, heart or metabolic disorders, fetal infections, clear teratogenic factors, Apgar at five minutes < 6 and grade III or IV hemorrhages), we see no way of re-analyzing our data in order to explain the relationship between birth order and yawning rates in light of a potential mediation of clinical variables. However, we have to point out that this finding is not the main result of our study (which pertains the association between hunger and yawning rates) and the excerpts the reviewer refers to does not in any way suggest that we have direct evidence of this difference being related to stress, but merely highlight that these results are consistent with this hypothesis, as second born twins have been found in previous studies to be prone to higher neonatal morbidity. In order to emphasize the hypothetical nature of this mediation, we introduced a sentence in the discussion (see below).

Q8: "Moreover, additional studies will be needed in order to directly test the hypothesis that stress and hunger-related modulations in neonates are in fact ascribable to cholinergic and ACTH-mediated pathways, as postulated by Collins and Eguibar [54]. In particular, further research is needed to confirm whether the effect of birth order on yawning rates in twins is due to stress-related factors associated with birth or to other variables."

Minor points

Introduction

Line 63 – eliminate “of”

RESPONSE: This sentence was removed and replaced by a more detailed explanation of the rationale for investigating yawning in preterm neonates, in response to reviewer #2's Q1.

Reviewer #2: Dear Authors,

Q1: you report here on a study examining factors that affect/modulate yawn frequencies in preterm neonates. Overall I think the study is well designed and executed, and the results are clear and comprehensible. As a results, some of your conclusions are therefore also justified. However, it remains a bit unclear what the actual rationale for this study was; i.e. why is it specifically interesting to study yawning in preterm neonates? What are the actual hypotheses. You summarize some of the existing hypotheses about yawning, but seem to convolute proximate and ultimate accounts and as a consequence present them as being mutually exclusive, which they in fact do not need to be. Nevertheless, the pletoria of hypotheses might be a reason to investigate further, yet the rationale to then look at pre-term neonates remains obscured, and without proper predictions at the end of the introduction the study does feel a bit like a fishing expedition. Therefore, I would like to see the study better embedded in the theoretical framework, creating a more balanced and useful account of your findings. Apart from that, I only have some minor suggestions/comments:

RESPONSE: We wish to thank the reviewer for their consideration of our work as well as for their comments. Based on both reviewers' as well as the editor's comments, we substantially revised the introduction, including a potential interpretation of our data based on the thermoregulatory feeding hypothesis (lines 89-94), as well as a more articulated explanation of the reasons for investigating yawning in the perinatal period (lines 67-78). We also specified the distinction between proximate and ultimate mechanisms involved in yawning modulation in order to make clear that not all theories about yawning are necessarily mutually exclusive (lines 58-64). Finally, we have to point out that investigating preterm neonates comes with serious methodological constraints, often translating to a difficulty in manipulating and controlling variables. This is mainly due to the fact that the policies and schedules of the Neonatal Intensive Care Unit (NICU) don't allow great flexibility in data collection, as neonates that are hospitalized in this context are extremely fragile.

Q2: l. 41: Please refer to Massen et al. 2021 Communications Biology, as this study is a much more elaborate account of the ubiquity of yawning in vertebrates (more species, more recent)

l. 49: If you want to refer to Apes, then please also cite van Berlo et al. 2020 Scientific Reports, which shows contagious yawning in Orangutans, since the current references only refer to homo sapiens and the two pan species

RESPONSE: We cited both studies.

Q3: l. 55: Not all of the hypothese above are functional ones, some are mechanistic and therefore, not all of these hypothese are mutually exclusive. Please make sure you have the framework right here, as it allows for a better understanding of where to place your results in the end.

RESPONSE: We added a paragraph explaining this issue, as explained in response to Q1 (see below):

"On the other hand, although these theories have been often presented as competing explanatory alternatives, several scholars have highlighted the need of distinguishing proximate and ultimate explanations when investigating mechanisms underlying the manifestation of yawning behavior [25]. In particular, advocates of the brain cooling hypothesis have argued that, while, e.g., empathy, familiarity or communication might serve as proximal mechanisms explaining yawning modulation, the ultimate function(s) of yawning is likely physiological and should be able to explain proximate mechanisms as well [26]."

Q4: l. 67: Please cite Gallup et al. 2021 Scientific Reports that also shows the effect of amount of sleep on yawn modulation

RESPONSE: We don't think this reference is pertinent here, as we did not investigate the effect of amount of sleep on yawn modulation. Moreover, preterm neonates in particular don't show circadian rhythms before one month corrected age (see e.g. Ivars et al., 2017, PLoS ONE, https://doi.org/10.1371/journal.pone.0182685). Also, Giganti et al., 2007 (https://doi.org/10.1016/j.infbeh.2007.03.005) investigated potential circadian variations in yawning frequencies of preterm neonates and found none.

Q5: l. 69-72: What is the hypothesis? Why should hunger lead to a modulation of yawning?

RESPONSE: See the response to Q1. In particular, besides being based on prior evidence in adults, the hypothesis of a hunger-related modulation of yawning resonates with the brain cooling theory of yawning as prior evidence shows that feeding in preterm neonates is integrated into a heat production episode.

l. 91: What are stress-related yawns? Please provide a reference, also to inform us about the hypothesis here, which is kind of missing.

Q6: We introduced two references with this regard (references 9 and 10).

Q7: l. 101 what does appropriate for gestational age mean? This is probably very familiar lingo for physicians, but you are reporting to a general journal, so please take into account the various backgrounds of your readers.

RESPONSE: We specified the definition of appropriate for gestational age (", i.e. whose birth weight was between the 10th and 90th percentiles for the infant’s gestational age and sex).

Q8: l. 114-122: The number of observation minutes seems rather low to me. Do you have any comparisons that allow us to gather how well such observational scheme represents the complete picture? and how prone it now is to random error?

RESPONSE: Considering that neonates were fed on average every 2/3 hours, a longer observation time would have compromised the relative homogeneity of the pre- and after-feeding conditions. Moreover, the analysis of spontaneous behavior in preterm neonates has been very often (if not almost always) performed on similar if not smaller observation times (see, e.g. Cioni, Prechtl, 1990, Early Hum Dev; Butcher et al., 2009, Child Psychology and Psychiatry). Finally, the fact that we actually found the expected differences (all with p<.001) is evidence of the fact that random error was not sufficient to obscure the investigated phenomena.

Q9: l.170. why did you use the natural log. of time for the offset and not just the actual time?

Because, in accordance with the count nature of our data and of a preliminary analysis of its distribution, we adopted Poisson regressions as analytic method, instead of calculating the rate of yawns per minute, we specified in our models the log-transformed value of the observation time as the offset (see also our response to reviewer #1's Q3). This is, to the best of our knowledge, the state-of-the-art approach when dealing with count data with varying exposures (see, e.g., Hutchinson & Holtman, 2005: https://doi.org/10.1002/nur.20093). In general, although it is somehow common practice to treat count data as interval/ratio variables, Poisson methods are preferable when dealing with skewed count data (Nussbaum et al., 2008: https://doi.org/10.4135/9781412995627.d26).

Q10: l.179. If there are no specific hypotheses, why would you reduce the power of your analyses (on a relatively small sample) by adding additional variables?

l.186. Similarly, why was gender included? Are there any specific hypotheses about gender effects?

RESPONSE: Because preterm neonates are in a very peculiar developmental position, controlling for age variables is crucial in order to exclude potential confounders as well as to explore possible developmental trends. Moreover, because yawning in preterm neonates has been largely neglected, checking for potential effects of sex and age variables serves an exploratory function that can serve as a guide for future studies.

Q11: l.2019-221. Please remove this sentence as it is redundant. There were no effects. p = 016 is not CLOSE to significance.

RESPONSE: We did not mean to suggest that it was: we had specified it in order to show that no effect was close to significance. However, we accept the reviewer's suggestion to delete that sentence.

Q12: l.246ff This conclusion is not justified as the relationship is indirect.

RESPONSE: We hope that the current framing of our study can be more convincing to the reviewer, once the rationale and hypotheses are more clear. Moreover, we explicitly recognized the fact that the relationship is indirect and could be explained by the brain cooling theory (lines 279.281) and we introduced a sentence in the discussion calling for future research to investigate the nature of these relationships (lines 306-309).

6. PLOS authors have the option to publish the peer review history of their article (what does this mean?). If published, this will include your full peer review and any attached files.

Do you want your identity to be public for this peer review? For information about this choice, including consent withdrawal, please see our Privacy Policy.

Reviewer #1: No

Reviewer #2: No

---

## [Decision Letter · Decision Letter 1]

22 Apr 2022

Factors affecting yawning frequencies in preterm neonates

PONE-D-22-01955R1

Dear Dr. Dondi,

We’re pleased to inform you that your manuscript has been judged scientifically suitable for publication and will be formally accepted for publication once it meets all outstanding technical requirements.

Kind regards,

Andrew C Gallup, Ph.D.

Academic Editor

PLOS ONE

Additional Editor Comments (optional):

Reviewers' comments:

Reviewer's Responses to Questions

**Comments to the Author**

1. If the authors have adequately addressed your comments raised in a previous round of review and you feel that this manuscript is now acceptable for publication, you may indicate that here to bypass the “Comments to the Author” section, enter your conflict of interest statement in the “Confidential to Editor” section, and submit your "Accept" recommendation.

Reviewer #1: All comments have been addressed

Reviewer #2: All comments have been addressed

2. Is the manuscript technically sound, and do the data support the conclusions?

Reviewer #1: Yes

Reviewer #2: Yes

3. Has the statistical analysis been performed appropriately and rigorously? 

Reviewer #1: Yes

Reviewer #2: Yes

4. Have the authors made all data underlying the findings in their manuscript fully available?

Reviewer #1: Yes

Reviewer #2: Yes

5. Is the manuscript presented in an intelligible fashion and written in standard English?

Reviewer #1: Yes

Reviewer #2: Yes

6. Review Comments to the Author

Reviewer #1: I am happy with the review provided by the authors. They better explained the rationale at the basis of their analyses.

Reviewer #2: I think you did a nice job addressing my comments and I am happy to recommend accepting your paper for publication.

7. PLOS authors have the option to publish the peer review history of their article (what does this mean?). If published, this will include your full peer review and any attached files.

Reviewer #1: No

Reviewer #2: No

---

## [Editor Report · Acceptance letter]

28 Apr 2022

PONE-D-22-01955R1 

Factors affecting yawning frequencies in preterm neonates 

Dear Dr. Dondi:

I'm pleased to inform you that your manuscript has been deemed suitable for publication in PLOS ONE. Congratulations! Your manuscript is now with our production department. 

Kind regards, 

on behalf of

Andrew C Gallup 

Academic Editor

PLOS ONE